# Frenkel-Poole Mechanism Unveils Black Diamond as Quasi-Epsilon-Near-Zero Surface

**DOI:** 10.3390/nano13020240

**Published:** 2023-01-05

**Authors:** Andrea Orsini, Daniele Barettin, Sara Pettinato, Stefano Salvatori, Riccardo Polini, Maria Cristina Rossi, Alessandro Bellucci, Eleonora Bolli, Marco Girolami, Matteo Mastellone, Stefano Orlando, Valerio Serpente, Veronica Valentini, Daniele Maria Trucchi

**Affiliations:** 1Università degli Studi Niccolò Cusano, “ATHENA” European University, Via don Carlo Gnocchi, 3, 00166 Roma, Italy; 2Istituto di Struttura della Materia, ISM-CNR, 00015 Monterotondo Stazione, Italy; 3Dipartimento di Scienze e Tecnologie Chimiche, Università degli Studi di Roma “Tor Vergata”, Via della Ricerca Scientifica, 00133 Rome, Italy; 4Department of Electronic Engineering, Università degli Studi di Roma Tre, Via Vito Volterra 62—Ex Vasca Navale, 00154 Roma, Italy

**Keywords:** diamond, LIPSS, Frenkel-Poole effect, ENZ

## Abstract

A recent innovation in diamond technology has been the development of the “black diamond” (BD), a material with very high optical absorption generated by processing the diamond surface with a femtosecond laser. In this work, we investigate the optical behavior of the BD samples to prove a near to zero dielectric permittivity in the high electric field condition, where the Frenkel-Poole (FP) effect takes place. Zero-epsilon materials (ENZ), which represent a singularity in optical materials, are expected to lead to remarkable developments in the fields of integrated photonic devices and optical interconnections. Such a result opens the route to the development of BD-based, novel, functional photonic devices.

## 1. Introduction

Since the first report in 2014 [1] about the enhanced optical absorption property in the visible range of the fs-laser treated diamond surface, considerable studies have been performed to analyze and optimize the treatments inducing surface nanostructuring (see Figure 1) [2,3]. This material, called “Black Diamond” (BD), is able to combine a very high optical absorptance, due to the formation of regular nanostructures over its surface, with a high mobility of the charge carriers [4], thus conceiving BD as a possible choice to photo-thermionic cathodes in high-temperature solar cells [5]. Furthermore, the possibility of deploying nanostructured diamonds for new applications as near-IR antireflection coatings [6,7] was demonstrated. On the other hand, this treatment introduces irregularities in the atomic arrangement of carbon atoms, and therefore the strain in its lattice, as it happens also in partially graphitized diamond substrates [8,9]. In optimized BD samples, where surface nanostructuring has a pseudo-bidimensional periodicity, an increased number of defects are present in the surface, as demonstrated by the increase of optical absorbance [10], and, correspondingly, the surface becomes orders of magnitude higher conductive than the bulk. Through thermal annealing up to 800 K and stress relaxation, it is possible to observe sheet resistance down to about 25 kΩ at high temperature and spintronic effects through the sequential current measurements in the cryogenic temperature range [11,12]. By enhancing the laser energy further, the spatial arrangement of the created defects percolation path breaks up and the black diamond surface becomes more insulating. On such highly defective BD samples it is possible to perform high electric field current measurements and observe FP conduction mechanism.

The application of large electric fields induces a considerable bending of the conduction band before reaching the breakdown condition. According to the Frenkel-Poole effect (FP), under a high electric field *E*, a trapped electron can jump to the conduction band without large thermal energy contributions. This induces a consequent measurable nonlinear enhancement of the current density *J*, following the equations:(1)J=J0T·eβE ,β=q32kBTπϵ0ϵr,  E=Vd
where, apart from well-known physical and mathematical constants (*k*_B_ the Boltzmann constant, *q* the electron charge and pi), and easily measurable experimental parameters (*T* the absolute temperature and *d* the electrodes distance), the only unknown parameter influencing the current/voltage relation is *ε_r_*, the material dielectric permittivity [13].

In this paper, we demonstrate through the FP effect that by enhancing the surface nanostructuring of a diamond (bulk relative permittivity equal to 5.5), it is possible to obtain its value diminution up to about 10^−2^. This is equivalent to considering the black diamond surface almost as an Epsilon Near Zero material (ENZ), representing a singularity in optical materials [14,15,16,17,18]. As observed for other ENZ surfaces [19,20,21], the coupling of such surfaces with nanoantennas allows us to obtain optical non-linearities and strongly improve the directionality of the emitted electromagnetic field; it is expected to lead to remarkable developments in the fields of integrated photonic devices and optical interconnections through the modulation of the Purcell effect [22] (i.e., the increase in spontaneous emission rate generally used to classify nanoantennas). Thanks to the presence of NV-centers, simple diamond nanoparticles also act as active dielectric nanoantennas with a high Purcell-factor [23]. Importantly, we note that NV-centers are also allocable by controlled surface laser writing [24], and it is, therefore, possible to combine in diamonds new electromagnetic effects to quantum properties of single photon emitters, making diamonds a leading contender as election material for the future quantum computer industry.

Furthermore, the demonstrated strong reduction of the dielectric polarizability through surface nanostructuring and the defects ordering on a wide band-gap material, like a diamond, in which n-type doping is physically very difficult, opens the way to a new procedure to realize the ENZ surface for photonic devices. In fact, ENZ materials have different fabrication techniques [25], but the most widely adopted and technologically convenient technique is to strongly enhance the free carrier concentration by heavy n-type doping of silicon or III-V semiconductors and to deploy the charges’ collective motion at plasma frequency [26,27,28]. We infer that highly defected surfaces with shallow electron traps may behave towards electromagnetic waves in a similar manner to the previously mentioned degenerated semiconductors. Even if this physical phenomenon should be confirmed by specific electromagnetic experiments, this is the first scientific report on quasi-ENZ effect on a diamond substrate up to our knowledge, and it can be useful to the large community of scientists working on diamond devices, from biosensors [29,30,31,32] to dosimeters [33,34] and to quantum computing [35,36]. In fact, as easy examples, the localized implementation of ENZ surfaces on diamonds gives the opportunity to improve optical interconnections between individual quantum building blocks (see Figure 2) or to improve nanoantennas radiation emission efficiency [19].

## 2. Materials and Methods

Polycrystalline diamond substrates with a lateral dimension of 10 × 10 mm^2^ and a thickness of 250 µm were purchased by Element Six Ltd. The sample surface was treated with a laser Spectra-Physics Spitfire Pro 100 F 1 K XP (pulse duration about 100 fs, wavelength 800 nm, repetition rate up to 1 kHz). The horizontally polarized laser beam was focused perpendicularly to the diamond plate surface in a high-vacuum chamber (pressure < 10^−7^ mbar) with a spot size on the sample of 150 µm diameter. The pulse energy was kept constant at 3.6 mJ/pulse, corresponding to a power density one hundred times larger than the diamond damage threshold (200 MW/µm^2^). The black diamond was created with double-nanotexturing with a total accumulated laser fluence of 6.25 kJ/cm^2^ on the diamond surface along the X-Y directions, and was obtained following the two-step bustrophedic procedure described in [37], by means of a computer-controlled micrometric X-Y translational stage. In the first step, a one-dimensional LIPSS structure was defined along the X direction. In the second step, the LIPSS structure was redefined along the Y direction thanks to the incubation effect guaranteeing a higher laser light absorption than the sample before the first treatment step. After the laser treatment, both samples were immersed in a strongly oxidizing solution (H_2_SO_4_:HClO_4_:HNO_3_ in the 1:1:1 ratio) for 15 min at the boiling point in order to remove any debris, as well as non-diamond carbon phases. Finally, the samples were cleaned by ultrasound sonication and rinsed in deionized water.

Surface morphology was studied by a Zeiss LEO Supra 35 field emission gun scanning electron microscope (FEG SEM). Secondary electron micrographs were acquired with an in-lens detector. The electron beam energy was set to 10 keV. 

Scanning Kelvin probe measurements of the central part of the black diamond sample were performed by an OmegaScope platform integrated into a LabRAM HR Evolution Raman microscope (HORIBA Ltd., Kyoto, Japan). Morphology and KPFM imaging were performed in tapping mode, setting the operational amplitude at 20 nm, and using a conductive Cr-Au coating silicon pyramidal tip (MikroMasch HQ: NSC14/Al BS, Wetzlar, Germany) with a characteristic radius of ~35 nm and about 110 kHz as the resonance frequency. The scan rate was fixed at 0.5 Hz. All the microscopies were acquired in kelvin probe mode and analyzed using the AIST-NT SPM control software. 

Raman spectroscopy was performed in back-scattering geometry by using a Dilor XY triple spectrometer (1 cm^−1^ resolution) equipped with an Ar+ laser (514.5 nm), a cooled CCD detector, and an adapted Olympus microscope arranged in confocal mode. The spot size was 2 mm, and 20 intensity points were collected to provide a suitable statistic. Line widths and intensities were determined using the Thermo-Grams Suite v9.2 software. 

After Raman spectroscopy experiments, a windowed stainless-steel sheet that was 50 µm thick was used as shadow-mask for thermal evaporation of Aluminum contacts on each sample. The distance between contacts was fixed at 1 mm for two parallel rectangles (see Figure 3).

Conductivity measurements up to 500 V in both bias voltage polarities were performed in a vacuum chamber with substrate temperature regulation performed by hot wires heating a copper hot-plate, on which the diamond plate is glued by silver paste to maximize the thermal flux. The current was acquired by a Keithley 487 Picoammeter/Voltage Source through two screwed molybdenum metal tips adhered in the central region of both Aluminum pads, both glued always by silver paste, by the application of a first ramp from 0 V → 500 V → 0 V (instrument maximum source voltage) and a second ramp with reversed electric field direction from 0 V → −500 V → 0 V.

## 3. Results

### 3.1. Morphological and Chemical Characterization

The sample analyzed in this paper is produced through a double-texturing technique with the target to maximize the largest total accumulated absorbed laser fluence, since the accumulated laser fluence of the second step was kept as large as the first: i.e., 50–50%. Indeed, the increase of absorptance after the first treatment is coupled with a decrease of the diamond ablation threshold. Therefore, the total accumulated absorbed laser fluence, defined as the product of the diamond absorptance at the laser wavelength and the delivered fluence, is characterized by a greater weight in the second treatment. As can be seen in Figure 4a, the diamond sample shows almost regular LIPSS, but is oriented according to the direction expected from the polarization direction of the second treatment, even if it is possible to observe perpendicular shallow lines deriving from the first treatment light polarization direction. Clearly, even if we did not perform an SEM session in the middle of the two steps of the BD fabrication, it is reasonable to assume that LIPSS formed horizontally after the first treatment, but thanks to the largest absorptance of the diamond surface during the second step, they were swept away during the formation of the new LIPSS in the second treatment. Traces of the LIPSS formed during the first step are evident in the inset zoom. The LIPSS periodicity related to the ratio of laser wavelength and the material refractive index does not appreciably change between the first texturing (First-step LIPSS) and the second one (Second-step LIPSS). A strong reduction of the dielectric permittivity may change the LIPSS periodicity, however:The ε_r_ change is supposed to be almost 2D. Therefore, it is unable to significantly contribute to the overall laser-diamond substrate interaction of the second texturing.The first treatment is performed at half of the standard power to produce a high-quality black-diamond surface. Therefore, the superficial ε_r_ change is also limited in value.The ε_r_ reduction is further enhanced by the two-stage texturing due to the incubation effect (see Appendix B).

In Figure 4b we show the Raman spectrum performed after the debris surface cleaning (green curve) showing considerable tensile stress in the diamond lattice. On the other hand, the annealing, performed at a temperature slightly higher than 500 °C for more than 24 h, revealed that high-temperature conditions influence reassembling of the diamond lattice and stress release in the surface layer. Therefore, it was decided to first bring the sample to the maximum temperature and second analyze its electrical behavior during the cooling phase (see Section 3.2).

A main point regarding black diamond nanostructured surface is the relation between the trap’s disposition and the periodicity of the laser-induced surface nanowaves. As stated in the introduction, black diamond samples with high ordering of the surface waves exhibit a superior conductivity on behalf of samples that absorbed even more laser energy but suffered a complete reorganization of the surface waves like the one analyzed in this paper. Therefore, it is reasonable to suppose that the surface waves are characterized by distinct traps depth levels. We performed surface Kelvin-Probe measurements, widely employed to study defects’ energy levels in organic electronic materials. Indeed, as shown in Figure 5, the higher black diamond zones of the laser-induced nanowaves, estimated to be at about 45°, have a distinctly more positive surface potential than the lower ones, with edge potentials reaching values of a hundred millivolts, which is more than twice the RT thermal voltage. Positive potentials mean that electrons are trapped in those regions [38,39] through localized defects with energy close to the conduction band.

### 3.2. Electrical Measurements

We performed high electric-field *J*(*E*) measurements (up to 5 kV/cm) with 23 steps of temperature in both the polarities of the applied electric field, with typical results of the first voltage ramp (0 V → 500 V) shown in Figure 6a–d (four I/V curves at different temperatures with relative Matlab fitting lines, see next paragraph), while opposite polarity electric field results are shown in Appendix A. Usually, it is possible to reveal the coefficient *β* of the FP effect by plotting the logarithmic of the current density (logJ) vs. the square root of the voltage (√*V*) and extracting its slope *S* = *β*/√*d*, where d is the distance between the measurement electrodes. However, in order to be more accurate, we decided to operate the coefficient extraction directly by using the Matlab “Curve Fitting” tool with the imposition of Equation (2) directly to the original data (see Figure 6). We noticed that the switching point between the two exponential regimes always falls in the 230 V–380 V range. Therefore, we applied three different fittings for each measurement:A.A Linear Fitting (I = A_Offset_ + G_Lin_*V), excluding point acquired at voltages > 100 V (red lines), where A_Offset_ is the little initial bias error of the instrument and G_Lin_ is the overall conductance of the sample at low voltages.B.A First Exponential Fitting (I = G_1_^FP^*V*exp(β_1_√V)), excluding the point acquired at voltages > 230 V (blue lines), where G_1_^FP^ represents the sample linear conductance to be combined with the exponential contribution and β_1_ represents the FP coefficient divided by the contact distance square root.C.A Second Exponential Fitting (I = G_2_^FP^*V*exp(β_2_√V)), considering all the experimental points (green lines), where G_2_^FP^ and β_2_ have the same meaning of G_1_^FP^ and β_1_.

It is possible to notice that the difference between the two exponential regimes changes with temperature. Such a difference is evident at room temperature (RT), it increases in the range 50 °C → 200 °C and it diminishes down to vanish at the highest available temperature, close to 500 °C. The extracted parameters (fits B and C of the previous list) are reported in Figure 7a (conductance) and Figure 7b (FP exponential coefficient) as a function of the 23 temperatures of measurement. Such results show that conductance increases exponentially with temperature as it is expected for thermal activation of charges trapped in shallow defects [40] and that the first exponential regime is characterized by a lower conductance value. In Figure 7b, it is instead evident that the exponential coefficient is slightly higher in first low-voltage range. On the other hand, the curves of Figure A2 regarding this factor show a reversed situation with cases of temperatures showing exactly the same exponential coefficient extracted but with values that always range from about 0.6 at the lowest temperatures to less than 0.3 at the highest ones.

## 4. Discussion

As expressed in Equation (1), the behavior of the current-electric field curve at a given temperature is determined only by the optical dielectric constant of the insulating film. In the vast majority of articles, the identification of the FP effect as the dominant current mechanism is made by plotting the sample conductance as a function of the square root of the applied electric field, leading to a straight line from which the value of the high-frequency dielectric constant is calculated [13]. In diamond, the frequency of the extracted *k* should correspond to the timing related to the fast electron transport in the conduction band (ε_r_~5.5) and not to the static dielectric response (ε_r_~7.8). In fact, in literature relative to the FP effect for diamond samples, it is typically found that a value for the exponential coefficient of about 3.2 × 10^−4^ eVcm/V, properly corresponding to an ε_r_ = 5.5 [41,42], even though higher values have been found under irradiation [43].

By substituting in Equation (1) the inherent physical constants, we can evaluate @ RT the following relation between the absolute value of dielectric permittivity and the parameter (β_1,2_) extracted by the I/V curves fitting:(2)$$\beta_{1,2}\left(\varepsilon_{r}=1,\;\mathrm{RT}\right)=92.7*10^{-3}\;\left[\frac{1}{\sqrt{V}}\right];\;\to \;\varepsilon_{r}\left(T\right)={\left(\frac{92.7*10^{-3}}{\beta_{1,2}}*\frac{300K}{T\left(K\right)}\right)}^{2};$$
where a larger measured slope corresponds to lower values of dielectric permittivity, according to the correction due to temperature variation.

If we consider the results shown in the previous section, we immediately understand that the exponential increase of the current with the electric field is much larger than the one calculated for Vacuum/Air. In fact, the largest value of *β*_1,2_, measured just close to RT, corresponds to a value of 0.6, almost one decade larger than the one corresponding at $\varepsilon_{r}=1$, and therefore the measured slope corresponds to a relative dielectric permittivity close to 10^−2^.

In Figure 8a we report the behavior of the slope extracted from the various I/V curves analyses as a function of the temperature at which the measurement was conducted. While the slopes shown in Figure 7b decrease with temperature, this does not correspond to a constant value of dielectric permittivity since the variation due to temperature according to Equation (2) is much larger. Therefore, as reported in the adjacent Figure 8b, the extracted value of the relative dielectric permittivity in reality halves to values close to the hundredth.

FP conduction was observed in the non-linear region at high voltages for many diamond samples regardless of deposition processes [44,45]. Amplification of the FP effect was obtained through X-ray irradiation in [43], where it was found for the exponential coefficient that a parabolic dependence on the density of occupied traps such to reach more than quadruple values in a specific electric field range (2 kV/cm → 7 kV/cm). Such behavior was attributed to the intense trap occupation by the promotion of electrons from valence bands, causing coulomb interaction between the trap centers with subsequent barrier lowering with an estimated mean distance (λ_traps_) of the electron-filled trap states of about 70 nm. The energy reduction due to the coulombic interaction of close traps additive to the electric field contribution may be related to their distance (a) with the following equation based on the Vollmann model [46]:(3)$$FP_{energy}\left(T\right)=q*\Phi_{th}{}^{2}*\beta_{1,2}{}^{2}*\frac{d}{a}\;;$$
where $\Phi_{th}$ is the thermal potential (see Figure 9). Since we are not observing a transition from Poole to Frenekl-Poole conduction regimes, it is not possible to extrapolate the real traps’ distance from the I/V curves, but we can consider that, given the high extrapolated beta coefficient, FP electron emission happens at distances of hundreds of nm, thus corresponding to the spatial gap between laser induced nanowaves (see Figure 4a and Figure 5b). Obviously, such value is just a tentative estimation and new, more complex transport experiments should be performed in order to correctly evaluate it. We can also note that the FP effect, as shown in Figure 8, increases with temperature, allowing us to estimate the lower values of the dielectric permittivity. Such behavior perfectly agrees with the increase of the conductance due to the thermal activation of electrons in the conduction band, which interacts at an increased rate with the empty trap center positively charged, and enhances the traps filling percentage at the origin of the FP effect reduction in BD.

## 5. Conclusions

Defected surfaces have plenty of new physics phenomena to be analyzed, even more if the defects’ geometry follows a specific regular pattern, as in the case of femtosecond laser surface-induced LIPSS. In this paper, we have shown that the electrical conduction of double nanotextured BD films is strongly influenced by the applied electric field. By applying the FP model to the I/V measurements, it was possible to reveal a dielectric permittivity near zero for the BD sample. The effect of the electric field over the trapped charges (FP effect) is appreciable just after it reaches 1 kV/cm, which is, to our knowledge, far lower than any other reported value over the presence of the FP effect in diamonds. The dielectric permittivity furthermore reduces its absolute value up to reach values lower than 0.01 for temperatures between 400 °C and 450 °C with electric fields lower than 2.3 kV/cm.

Such a result demonstrates that it is possible to reach quasi-ENZ polarizability conditions on diamond surfaces thanks to laser surface nanostructuring, an approach that is very likely to be extendable to other materials’ surfaces, widening the class of ENZ materials. This new property found on diamonds demonstrates again that it should be considered even more than other materials for the continued developments of novel functional photonic devices, since its properties are unique in nature and it may combine very low surface polarizability to other quantum properties related to its optically-active defect centers.

## Figures and Tables

**Figure 1 nanomaterials-13-00240-f001:**
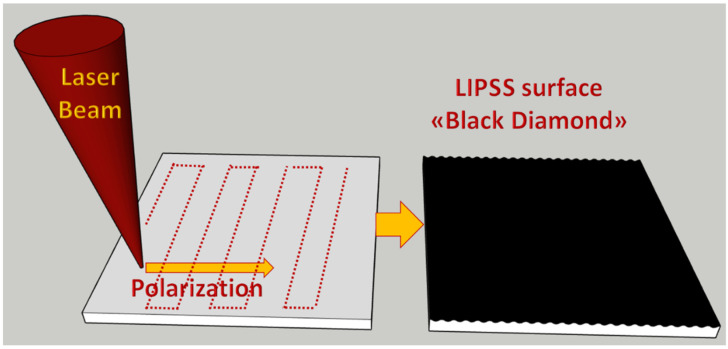
Schematic of the “Black Diamond” fabrication process.

**Figure 2 nanomaterials-13-00240-f002:**
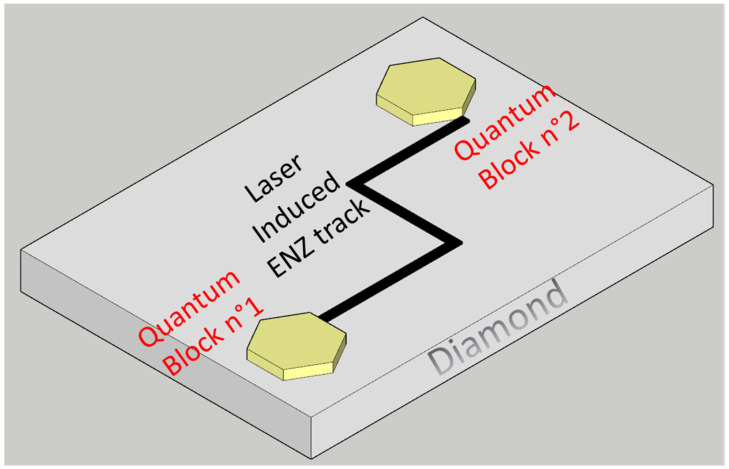
Sketch of the possible combination of quantum blocks on diamond substrates through spatially controlled laser induced ENZ waveguides.

**Figure 3 nanomaterials-13-00240-f003:**
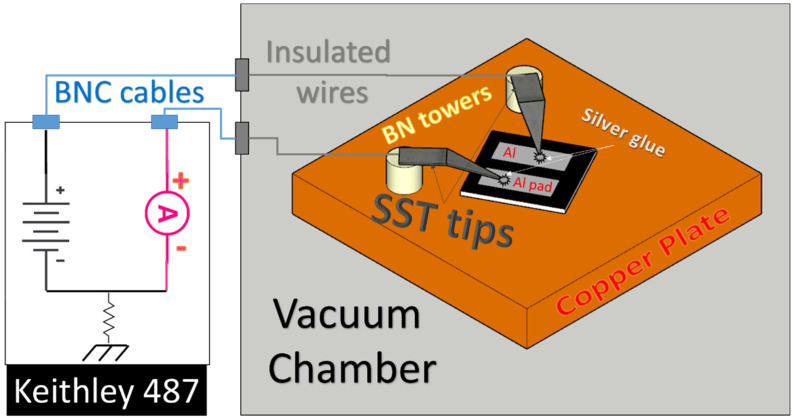
Sketch of the current measurements experimental setup.

**Figure 4 nanomaterials-13-00240-f004:**
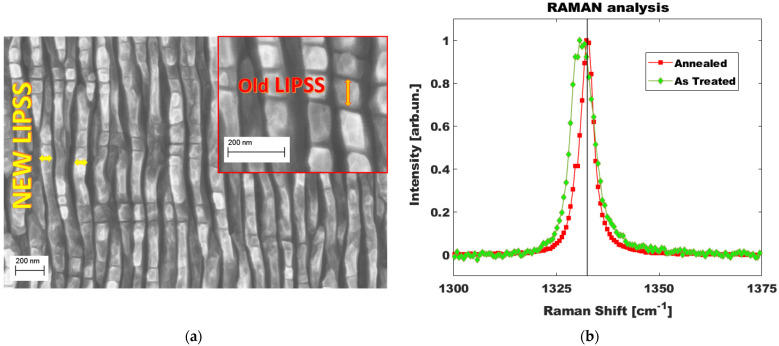
Surface characterization of black diamond samples (**a**) SEM image of the surface; (**b**) Raman analysis of the diamond peak at 1331.2 cm^−1^ before (green curve) and after (red curve) of the annealing at temperature slightly higher than 500 °C for more than 24 h. Black line corresponds to natural diamond Raman peak.

**Figure 5 nanomaterials-13-00240-f005:**
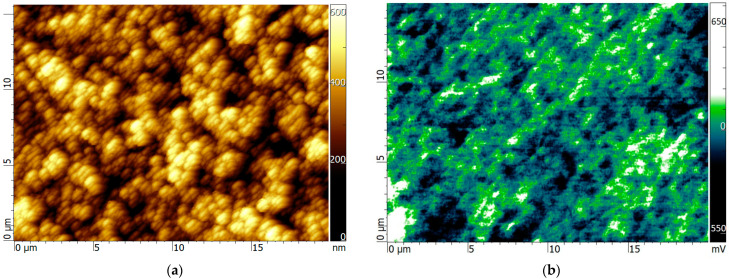
Surface potential maps of black diamond laser induced periodic surface nanostructures and their role as electron traps. (**a**) AFM image of the surface; (**b**) SKPM surface potential images.

**Figure 6 nanomaterials-13-00240-f006:**
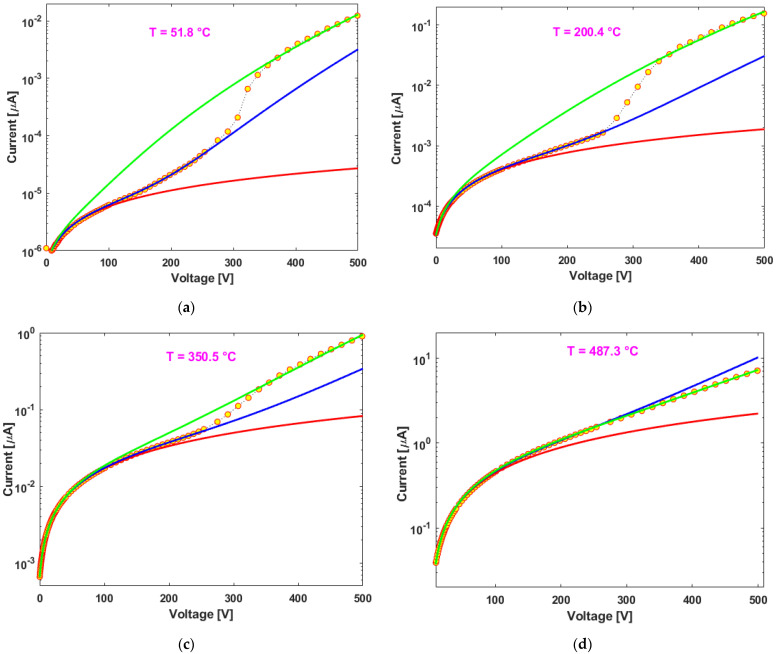
I/V curves relative to the positive voltage ramp with corresponding Matlab fitting for the linear part (red line), the first exponential part up to 230V (blue curve) and the total exponential behavior up to maximum voltage of 500 V (green curve). (**a**) measurement at about 50 °C; (**b**) measurement at about 200 °C; (**c**) measurement at about 350 °C; (**d**) measurement at about 500 °C.

**Figure 7 nanomaterials-13-00240-f007:**
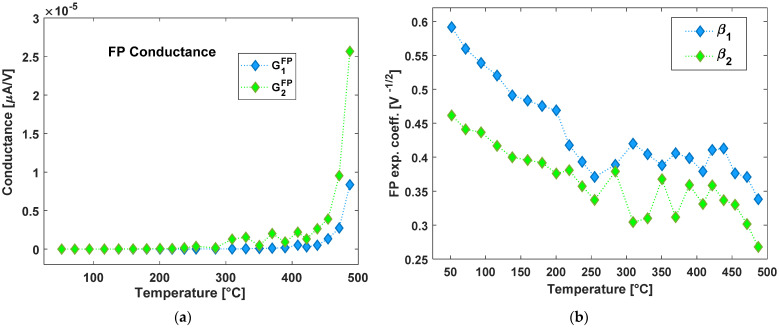
Frenkel-Poole- parameters as extracted from the I/V curves: (**a**) conductance for *V* < 230 V (blue curve, number 1 subscript in the legend) and for *V* < 500 V (green curve, number 2 subscript in the legend); (**b**) Extraction of the exponential coefficient for *V* < 230 V (blue curve, number 1 subscript in the legend) and for *V* < 500 V (green curve, number 2 subscript in the legend).

**Figure 8 nanomaterials-13-00240-f008:**
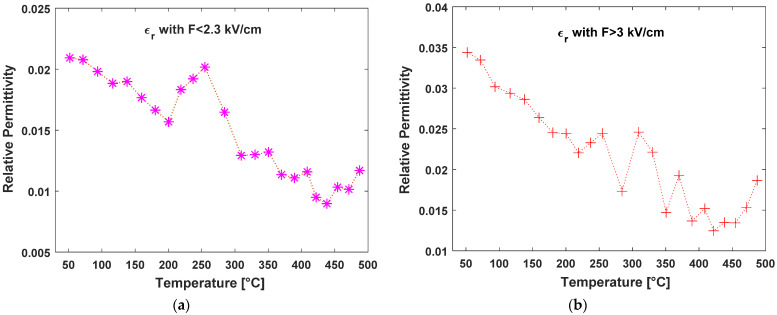
Relative Dielectric Permittivity as extracted by the I/V curves reported in Figure 6. (**a**) Extraction of parameter with V < 230 V; (**b**) Extraction of parameter with V > 300 V.

**Figure 9 nanomaterials-13-00240-f009:**
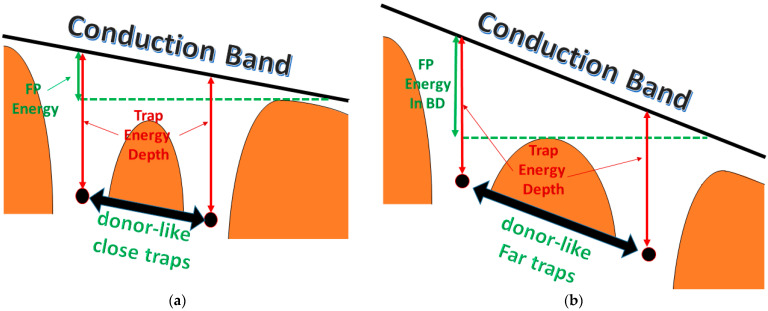
(**a**) FP energy in standard materials with high density filled traps with reciprocal coulombic interaction with highlighted barrier lowering. (**b**) FP energy in quasi-ENZ BD surface.

## Data Availability

The data presented in the current work are available on request from the corresponding authors.

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
