# Peer review of "Frenkel-Poole Mechanism Unveils Black Diamond as Quasi-Epsilon-Near-Zero Surface"

_nanomaterials, 2023, doi:10.3390/nano13020240_

Round 1

Reviewer 1 Report

Manuscript Number: nanomaterials-2135173-peer-review-v1

In this work, we investigate the optical behavior of the BD samples to prove a near to zero dielectric permittivity in the high electric field condition, where Frenkel-Poole effect takes place. Before publication, the author should make the following modifications.

1.       The author should highlight the connection between the manuscript and nanomaterials. From the current manuscript, there is still no effective connection.

2.       Introduction: The author should introduce the latest research progress in this field. The references in the current manuscript are too old.

3.       Some sentences have grammatical problems. I suggest the author revise them carefully.

4.       Two articles are helpful to improve the quality of manuscripts. (10.1016/j.jhazmat.2020.124889 and 10.1021/acsami.1c22035).

5.       Please explain in the manuscript what the study has implications for readers.

6.       Many formulas have appeared in previous studies. I suggest the author simplify the introduction of formulas.

Author Response

In this work, we investigate the optical behavior of the BD samples to prove a near to zero dielectric permittivity in the high electric field condition, where Frenkel-Poole effect takes place. Before publication, the author should make the following modifications.

  1. The author should highlight the connection between the manuscript and nanomaterials. From the current manuscript, there is still no effective connection.

We are sorry we did no highlight enough the connection between the surface nanostructuring and the measured properties. Now we added a new experiment in the morphological characterization section highlighting the link between the created surface nanowaves and the electron traps originating the peculiar surface properties.

  1. Introduction: The author should introduce the latest research progress in this field. The references in the current manuscript are too old.

We changed all the introduction improving the link between the different concepts we wanted to introduce and the latest research progress in this field adding the following new citations:

  1. Wu, J.; Xie, Z.T.; Sha, Y.; Fu, H.Y.; Li, Q. Epsilon-near-zero photonics: infinite potentials. Photonics Res. 2021, 9, 1616, doi:10.1364/prj.427246.
  2. Kinsey, N.; DeVault, C.; Boltasseva, A.; Shalaev, V.M. Near-zero-index materials for photonics. Nat. Rev. Mater. 2019, 4, 742–760, doi:10.1038/s41578-019-0133-0.
  3. Reshef, O.; De Leon, I.; Alam, M.Z.; Boyd, R.W. Nonlinear optical effects in epsilon-near-zero media. Nat. Rev. Mater. 2019, 4, 535–551, doi:10.1038/s41578-019-0120-5.
  4. Niu, X.; Hu, X.; Chu, S.; Gong, Q. Epsilon-Near-Zero Photonics: A New Platform for Integrated Devices. Adv. Opt. Mater. 2018, 6, 1701292, doi:https://doi.org/10.1002/adom.201701292.
  5. Pasquarelli, A.; Andrilli, L.H.S.; Bolean, M.; Reis Ferreira, C.; Cruz, M.A.E.; de Oliveira, F.A.; Ramos, A.P.; Millán, J.L.; Bottini, M.; Ciancaglini, P. Ultrasensitive Diamond Microelectrode Application in the Detection of Ca2+ Transport by AnnexinA5-Containing Nanostructured Liposomes. Biosensors 2022, 12, 1–15, doi:10.3390/bios12070525.
  6. Xie, H.; Li, X.; Luo, G.; Niu, Y.; Zou, R.; Yin, C.; Huang, S.; Sun, W.; Li, G. Nano-diamond modified electrode for the investigation on direct electrochemistry and electrocatalytic behavior of myoglobin. Diam. Relat. Mater. 2019, 97, 107453, doi:https://doi.org/10.1016/j.diamond.2019.107453.
  7. Feng, Z.; Gao, N.; Liu, J.; Li, H. Boron-doped diamond electrochemical aptasensors for trace aflatoxin B1 detection. Anal. Chim. Acta 2020, 1122, 70–75, doi:https://doi.org/10.1016/j.aca.2020.04.062.
  8. Yang, K.-H.; Narayan, R.J. Biocompatibility and functionalization of diamond for neural applications. Curr. Opin. Biomed. Eng. 2019, 10, 60–68, doi:https://doi.org/10.1016/j.cobme.2019.03.002.
  9. Pettinato, S.; Girolami, M.; Olivieri, R.; Stravato, A.; Caruso, C.; Salvatori, S. Time-Resolved Dosimetry of Pulsed Photon Beams for Radiotherapy Based on Diamond Detector. IEEE Sens. J. 2022, 22, 12348–12356, doi:10.1109/JSEN.2022.3173892.
  10. Pettinato, S.; Orsini, A.; Girolami, M.; Trucchi, M.D.; Rossi, C.M.; Salvatori, S. A High-Precision Gated Integrator for Repetitive Pulsed Signals Acquisition. Electron. 2019, 8.
  11. Evans, R.E.; Bhaskar, M.K.; Sukachev, D.D.; Nguyen, C.T.; Sipahigil, A.; Burek, M.J.; Machielse, B.; Zhang, G.H.; Zibrov, A.S.; Bielejec, E.; et al. Photon-mediated interactions between quantum emitters in a diamond nanocavity. Science (80-. ). 2018, 362, 662–665, doi:10.1126/science.aau4691.
  12. Liu, K.; Zhang, S.; Ralchenko, V.; Qiao, P.; Zhao, J.; Shu, G.; Yang, L.; Han, J.; Dai, B.; Zhu, J. Tailoring of Typical Color Centers in Diamond for Photonics. Adv. Mater. 2021, 33, 2000891, doi:https://doi.org/10.1002/adma.202000891.

  1. Some sentences have grammatical problems. I suggest the author revise them carefully.

We revised all the paper and we highlighted in yellow in the resubmitted version all the introduced text changes.

  1. Two articles are helpful to improve the quality of manuscripts. (10.1016/j.jhazmat.2020.124889 and 10.1021/acsami.1c22035).

We read and appreciated the two suggested papers, especially the second one that we will introduce in a paper dedicated to LDH, to be submitted.

  1. Please explain in the manuscript what the study has implications for readers.

We added the following text evidencing as this paper may have strong implications in nanophotonic devices fabrication on diamond and new Fig. 2:

“Furthermore, the demonstrated strong reduction of the dielectric polarizability through surface nanostructuring and defects ordering on a wide band-gap material like diamond which n-type doping is physically very difficult, opens the way to a new procedure to realize ENZ surface for photonic devices. In fact, ENZ materials have different fabrication procedure [25], but the most technologically convenient is to strongly enhance the free carrier concentration by heavy n-type doping of semiconductors (silicon or III-V) and to deploy charges collective motion at plasma frequency [26–28]. Up to our knowledge this is the first scientific report on ENZ effect on a diamond substrate and it can be useful to all the large community of scientists working on diamond devices, from biosensors [29–32] to dosimeters [33,34] and finally quantum computing [35,36] by giving the opportunity to improve optical interconnection between individual quantum building blocks (see Fig. 2) but also to improve nanoantennas radiation emission efficiency [19].”

  1. Many formulas have appeared in previous studies. I suggest the author simplify the introduction of formulas.

We are sorry but, as reviewer n3 asked us to go more into detail with the model, we kept 3 equations as a total number. However, we removed one of the three originally present equations. We stress that even if equations are old, general readers may not know them and there are multiple definition of FP constants, i.e. w/ or w/o the thermal energy contribution, therefore it is important to express the assumed one.

We thank the reviewer for his precious comments, in the attachment the accordingly modified paper.

Reviewer 2 Report

The manuscript is dedicated to novel aspects in the production of black diamond-based metamaterials with potential applications in optical data processing, including for the development of metadevices such as quantum computers. The authors have developed methods to: (i) generate double-texturized black-diamond surfaces on polycrystalline diamond substrates, (ii) characterize the morphologic and electric properties of the resulted composite, and (iii) evaluate the behavior of epsilon-near-zero type of the simple metadevice they produced in order to investigate mechanisms of energy transfer in the conduction band of the resulted metamaterial.

The strong point of the work is related to the potential applications of the techniques that authors developed, provided that experimental refinements will be done in an attempt to enhance the reproducibility of the nano-texturing of the polycrystalline diamond substrata. An interesting (but incipient and poorly explained) model of the electrons promotion under thermal excitation was also developed.

However, in my opinion, a drawback of the manuscript consists in its ellipticity, and its telegraphic style. Obviously, the manuscript was written having in mind an exclusivist group of specialists, because few and poor explanations were inserted regarding the backgrounds of the scientific ideas and the "philosophy" of the experiments.

Moreover, the manuscript seems to have been written in a hurry. As arguments, please see rows No. 33, 46, 47, 217, 259, 322, two equations numbered as (3), and other more.

Technical comments

C1. To be more specific, please use "accumulated laser fluence" in row 78.

C2. In "Materials and Methods" section, please also describe the Raman investigation you have performed.

C3. Even if not mandatory, please insert an image of the junction you used to connect the silver electrodes (in Figure 2) with the voltage source, in order to evaluate/understand the contact parameters and quality. A schematic of the measuring device you constructed may be instructive too.

Recommendations

R1. Please enhance the clarity and completeness of the introduction section.

R2. Please insert more explanations regarding the model in Figure 7, which represents a potential scientific contribution of your work.

Author Response

The manuscript is dedicated to novel aspects in the production of black diamond-based metamaterials with potential applications in optical data processing, including for the development of metadevices such as quantum computers. The authors have developed methods to: (i) generate double-texturized black-diamond surfaces on polycrystalline diamond substrates, (ii) characterize the morphologic and electric properties of the resulted composite, and (iii) evaluate the behavior of epsilon-near-zero type of the simple metadevice they produced in order to investigate mechanisms of energy transfer in the conduction band of the resulted metamaterial.

The strong point of the work is related to the potential applications of the techniques that authors developed, provided that experimental refinements will be done in an attempt to enhance the reproducibility of the nano-texturing of the polycrystalline diamond substrata. An interesting (but incipient and poorly explained) model of the electrons promotion under thermal excitation was also developed.

However, in my opinion, a drawback of the manuscript consists in its ellipticity, and its telegraphic style. Obviously, the manuscript was written having in mind an exclusivist group of specialists, because few and poor explanations were inserted regarding the backgrounds of the scientific ideas and the "philosophy" of the experiments.

Moreover, the manuscript seems to have been written in a hurry. As arguments, please see rows No. 33, 46, 47, 217, 259, 322, two equations numbered as (3), and other more.

We thanks the reviewer for his detailed comments that we greatly appreciated. We checked and corrected the aforesaid rows and the numbering error.

Technical comments

C1. To be more specific, please use "accumulated laser fluence" in row 78.

We changed accordingly line 78.

C2. In "Materials and Methods" section, please also describe the Raman investigation you have performed.

We explained the Raman investigation in lines 90-95:

“Raman spectroscopy was performed in back-scattering geometry, by using a Dilor XY triple spectrometer (1 cm-1 resolution), equipped with an Ar+ laser (514.5 nm), cooled CCD detector and an adapted Olympus microscope arranged in confocal mode. The spot size was 2 mm and 20 intensity points were collected to provide a suitable statistic. Line widths and intensities were determined using the Thermo-Grams Suite v9.2 software.”

C3. Even if not mandatory, please insert an image of the junction you used to connect the silver electrodes (in Figure 2) with the voltage source, in order to evaluate/understand the contact parameters and quality. A schematic of the measuring device you constructed may be instructive too.

We changed Fig. 2 (now Fig. 3) adding the stainless steel metal tips and we evidenced that the contacts are made by using silver paste onto Aluminium pads. Previously they were erroneously tagged as silver. We also added the circuit of the external electrometer used for the high electric field current measurements.

Recommendations

R1. Please enhance the clarity and completeness of the introduction section.

We strongly changed all the introduction improving the link between the different concepts we wanted to introduce.

R2. Please insert more explanations regarding the model in Figure 7, which represents a potential scientific contribution of your work.

We improved Fig. 7 and we added relative equation with inverse proportionality between energy and traps distance. However, we noticed a my personal error applying proportionality in reverse way… so the value of FP emission distance is likely to be even hundreds of nm in agreement with the increased beta value.

“The energy reduction due to coulombic interaction of close traps additive to the electric field contribution may be related to their distance (a) with the following equation based on Vollmann model [46]:

(3)

where  is the thermal potential (see Fig. 9). Since we are not observing a transition from Poole to Frenekl-Poole conduction regimes it is not possible to extrapolate the real traps distance from the I/V curves but we can consider that given the high extrapolated beta coefficient, FP electron emission happens at distances of hundreds of nm thus corresponding to the spatial gap between laser induced nanowaves (see Fig. 4a and 5b).”

We thank the reviewer for his comments, especially for this last one that stimulated us to remove the error. In the attachment the accordingly modified paper.

Reviewer 3 Report

In this manuscript, the authors conducted the research on Frenkel-Poole mechanism unveils black diamond as Epsilon-2 Near-Zero surface.

There are a few comments as follows:

1. Introduction part is needed to be improved.

2. The rationale of research must be included in graphically in the last paragraph of introduction section. 

3. More characterization techniques can be included in the manuscript. 

Author Response

In this manuscript, the authors conducted the research on Frenkel-Poole mechanism unveils black diamond as Epsilon-2 Near-Zero surface.

There are a few comments as follows:

  1. Introduction part is needed to be improved.

We strongly changed all the introduction improving the link between the different concepts we wanted to introduce.

  1. The rationale of research must be included in graphically in the last paragraph of introduction section. 

We added a new figure 2 relative to the research rationale in aforesaid section.

  1. More characterization techniques can be included in the manuscript. 

We added a new characterization technique: SKPM in section 3.1.

We thank the reviewer for his comments, especially for this last one that stimulated us for new experiments..In the attachment the accordingly modified paper.
